# All-atom inverse protein folding through discrete flow matching

Kai Yi [* 1]   Kiarash Jamali [* 1]   Sjors H. W. Scheres [1]

## Abstract

The recent breakthrough of AlphaFold3 in modeling complex biomolecular interactions, including those between proteins and ligands, nucleotides, or metal ions, creates new opportunities for protein design. In so-called inverse protein folding, the objective is to find a sequence of amino acids that adopts a target protein structure. Many inverse folding methods struggle to predict sequences for complexes that contain non-protein components, and perform poorly with complexes that adopt multiple structural states. To address these challenges, we present ADFLIP (All-atom Discrete FLow matching Inverse Protein folding), a generative model based on discrete flow-matching for designing protein sequences conditioned on all-atom structural contexts. ADFLIP progressively incorporates predicted amino acid side chains as structural context during sequence generation and enables the design of dynamic protein complexes through ensemble sampling across multiple structural states. Furthermore, ADFLIP implements training-free classifier guidance sampling, which allows the incorporation of arbitrary pre-trained models to optimise the designed sequence for desired protein properties. We evaluated the performance of ADFLIP on protein complexes with small-molecule ligands, nucleotides, or metal ions, including dynamic complexes for which structure ensembles were determined by nuclear magnetic resonance (NMR). Our model achieves state-of-the-art performance in single-structure and multi-structure inverse folding tasks, demonstrating excellent potential for all-atom protein design. The code is available at https://github.com/ykiiiiii/ADFLIP.

*Equal contribution  [1]MRC Laboratory of Molecular Biology, Cambridge, UK. Correspondence to: Sjors H. W. Scheres <scheres@mrc-lmb.cam.ac.uk>.

*Proceedings of the $42^{st}$ International Conference on Machine Learning*, Vancouver, Canada. PMLR 267, 2025. Copyright 2025 by the author(s).

## 1. Introduction

The introduction of AlphaFold2 (Jumper et al., 2021) represented a breakthrough in the prediction of three-dimensional protein structures from amino acid sequence. AlphaFold2 also catalysed advances in *de novo* protein design, where one aims to design an amino acid sequence that adopts a given three-dimensional structure, in a process called inverse protein folding (Dauparas et al., 2022). Combined with diffusion-based methods to generate protein structures (Watson et al., 2023), inverse folding methods have opened new frontiers in the design of proteins with specific structural and functional properties, with profound implications for enzyme engineering (Lauko et al., 2024), antibody development (Bennett et al., 2024), and therapeutic interventions (Glögl et al., 2024).

Because AlphaFold2 only considered proteins, most protein design efforts to date have also focused on structures that exclusively comprise proteins. However, many biological reactions that are necessary to sustain life employ a much wider range of chemistry. For example, chemical modification of the amino acids that make up proteins may be used to signal distinct functional states; the binding of small-molecule ligands in specific protein pockets often affects function; many proteins interact with nucleic acids that carry genetic information; and many chemical reactions are catalysed by metal ions that are bound to enzymes. Recently, AlphaFold3 (Abramson et al., 2024) expanded the prediction of biological structures beyond proteins by modelling complex assemblies that also include nucleic acids, small molecules, metal ions, and chemical modifications of amino acids. This advance in structure prediction again created parallel opportunities for inverse folding (Dauparas et al., 2023), but the design of amino acid sequences for biomolecular complexes that contain a wide range of chemical entities remains more challenging than the design of proteins only.

An additional complication in protein design is that many biomolecular complexes are also structurally dynamic. Much like machines in daily life, the functional cycle of such complexes involves multiple structural states, even though their amino acid sequence is fixed. Although current models for inverse folding perform relatively well on static protein structures, they often struggle with complexes that

adopt multiple structural states, e.g. dynamic proteins that are studied by solution-state nuclear magnetic resonance (NMR) (Nikolaev et al., 2024; Yi et al., 2024).

In this paper, we present ADFLIP: a discrete flow-matching model for generating protein sequences based on the structure of bio-molecular complexes that can involve non-protein elements. As shown in Figure 1, ADFLIP generates sequences by jointly considering protein backbone structure and non-protein elements, such as nucleotides, ligands, or metal ions, while progressively incorporating predicted amino acid side chains as structural context during sequence generation. Because amino acid side chains often provide the chemical entities that define specific interactions with other molecules, considering the full information of the side chains is crucial for designing protein-ligand interactions. Our approach employs a multi-scale graph neural network as the denoising backbone, integrating both atom and residue-level information. Our iterative flow-based sampling framework enables ensemble sampling across different structural states, and it facilitates the integration of guidance signals to steer sample generation toward desired outcomes. The resulting model allows the design of biomolecular complexes with diverse chemical entities that adopt multiple conformational states and with defined protein properties.

## 2. Related work

**Inverse protein folding**   Predicting a sequence of amino acids that folds to a given structure using machine learning approaches was pioneered in (O'Connell et al., 2018; Wang et al., 2018), but the current paradigm of using graph neural networks (Dauparas et al., 2022; Hsu et al., 2022; Gao et al., 2023b; Yi et al., 2024; Zhu et al., 2024) was first proposed in (Ingraham et al., 2019). These methods typically encode the protein backbone as a graph and use graph neural networks to extract geometric information. Recently, SurfPro (Song et al., 2024) extended this approach by incorporating desired surface properties into the sequence design. Other methods, like LMdesign (Zheng et al., 2023) and KWdesign (Gao et al., 2023a), have leveraged both geometric information and information from evolutionarily related protein homologues through the combined use of structural information with protein language models. Saport (Su et al., 2024) introduced a structure-aware vocabulary to encode protein structures as discrete tokens, enabling general protein language models to generate sequences directly from structures. While most existing methods focus on protein-only backbone structures, LigandMPNN (Dauparas et al., 2023) broadened the scope of inverse folding by considering all-atom structures that also contain non-protein components like ligands, ions, and nucleotides.

**Discrete generative model**   Because mapping a protein's structure to its sequence represents a one-to-many mapping task, the sampling strategy for sequence generation presents an important design choice. Protein sequences, unlike natural language, do not have an inherent left-to-right ordering. Thus, although autoregressive generation (Ingraham et al., 2019; Hsu et al., 2022) was initially the dominant approach, various alternative sampling strategies have been proposed. ProteinMPNN (Dauparas et al., 2022) employs random ordering for sequence generation, while PiFold (Gao et al., 2023b) predicts amino acid probabilities in one step with independent sampling at each position.

Discrete generative models have undergone major recent developments. Discrete diffusion models were first introduced by Austin et al. (2021) and Hoogeboom et al. (2021), enabling diffusion-based generation of categorical data through forward corruption processes with Markov transition kernels. Campbell et al. (2022) extended this to continuous time by formulating discrete diffusion through Continuous Time Markov Chains. Recently, two independent works by Campbell et al. (2024) and Gat et al. (2024) proposed discrete flow matching, providing a simpler framework that allows faster training convergence. Here, we follow the discrete flow matching setting proposed in Campbell et al. (2024).

## 3. Methods

ADFLIP is a generative model $p_\theta(\boldsymbol{S}|\boldsymbol{X})$, with parameters $\theta$, for protein sequence design that samples discrete protein sequences $\boldsymbol{S} \in \{1, ..., 20\}^L$ conditioned on protein-complex structural inputs $\boldsymbol{X} = \{\boldsymbol{X}_{\text{protein}}, \boldsymbol{X}_{\text{non-protein}}\}$, where $\boldsymbol{X}_{\text{protein}} \in \mathbb{R}^{L \times 4 \times 3}$ represents the protein backbone coordinates of atoms $(N, C_\alpha, C, O)$, and $\boldsymbol{X}_{\text{non-protein}} \in \mathbb{R}^{m \times 3}$ is the non-protein structure components with $m$ atoms, $L$ is the length of protein sequence. During sampling, ADFLIP progressively incorporates side chain information $\chi \in \mathbb{R}^{L \times 4}$ while generating the sequence. ADFLIP consists of three key components: a discrete flow model for protein sequence generation; a multi-scale graph neural network denoiser that captures both residue-level and atom-level information; and a property guidance mechanism that enables controlled generation without model retraining.

### 3.1. Discrete flow model all-atom inverse folding

We construct the sequence generating probability flow $p_t(\mathbf{s}_t)$ at time $t$ that interpolates from a noise distribution $p_0(\mathbf{s}_0)$ to the data distribution $p_1(\mathbf{s}_1) = p_{\text{data}}(\mathbf{s}_1)$. Since modelling $p_t$ directly is complex, we instead define it through a simpler conditional flow $p_{t|1}(\cdot|\mathbf{s}_1)$ that interpolates from noise to the target sequence. This conditional flow has a closed form:

$$p_{t|1}(\mathbf{s}_t|\mathbf{s}_1) = \text{Cat}(t\delta\{\mathbf{s}_1, \mathbf{s}_t\} + (1-t)\delta\{\mathbb{m}, \mathbf{s}_t\}),$$

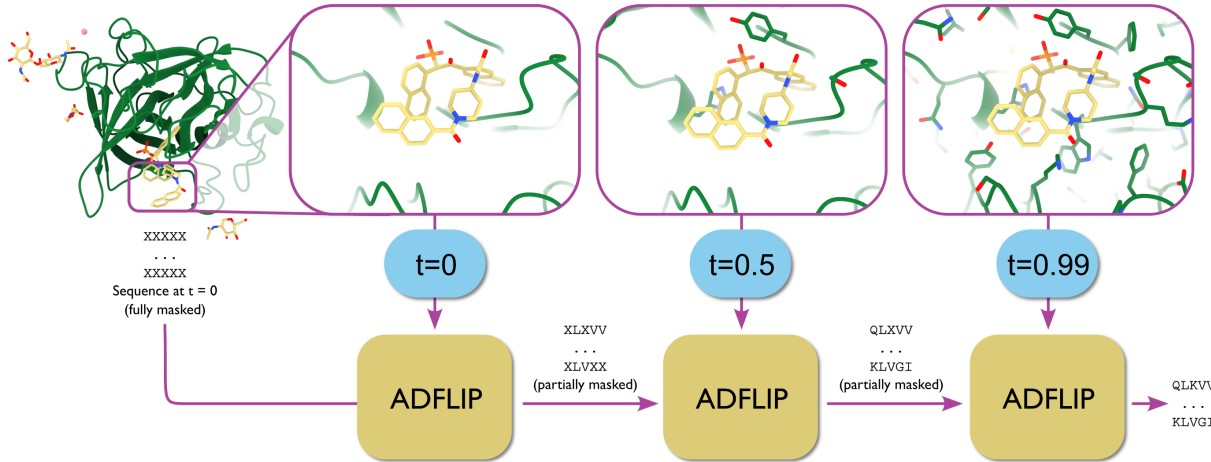

*Figure 1.* Overview of ADFLIP. ADFLIP is a flow matching generative framework for protein sequence design. Starting from $t = 0$, it progressively generates the protein sequence through sequential denoising using only the protein backbone and non-protein structure as inputs. The process begins with all amino acid positions masked (X) and gradually unmasks them to reveal the sequence. As amino acids are sampled, their corresponding side chain structures are predicted, providing additional structural context to guide subsequent denoising steps.

where $\mathbb{m}$ represents a mask token, $t \in [0, 1]$ is the time parameter, Cat is the categorical distribution, and $\delta\{i, j\}$ is the delta function. At $t = 0$, all positions are masked, while at $t = 1$, the sequence matches the target $\mathbf{s}_1$. With the conditional flow model, we define the marginal distribution $p_t$ by taking the expectation over the data distribution:

$$p_t(\mathbf{s}_t) = \mathbb{E}_{p_{\text{data}}(\mathbf{s}_1)}[p_{t|1}(\mathbf{s}_t|\mathbf{s}_1)].$$

To sample a sequence from $p(\mathbf{s})$, we need access to a rate matrix $R_t$ that defines the frequency and destination of state transitions, with the constraint that its off-diagonal elements are non-negative. The probability that state $\mathbf{x}_t$ will jump to a different state $j$ for the next infinitesimal time step $dt$ is $R_t(\mathbf{s}_t, j)dt$. The transition probability can be written as:

$$p_{t+dt|t}(j|\mathbf{s}_t) = \delta\{\mathbf{s}_t, j\} + R_t(\mathbf{s}_t, j)dt.$$

In practice, we simulate the sequence trajectory with finite time intervals $\Delta t$ using Euler steps:

$$\mathbf{s}_{t+\Delta t} \sim \text{Cat}(\delta\{\mathbf{s}_t, \mathbf{s}_{t+\Delta t}\} + R_t(\mathbf{s}_t, \mathbf{s}_{t+\Delta t})\Delta t). \quad (1)$$

We represent the rate matrix $R_t(\mathbf{s}_t, j)$ as an expectation over a simpler conditional rate matrix:

$$R_t(\mathbf{s}_t, j) = \mathbb{E}_{p_{1|t}(\mathbf{s}_1|\mathbf{s}_t)}[R_t(\mathbf{s}_t, j|\mathbf{s}_1)],$$

where the conditional rate matrix has the form: $R_t(\mathbf{s}_t, j|\mathbf{s}_1) = \frac{\delta\{\mathbf{s}_1, j\}\delta\{\mathbf{s}_t, \mathbb{m}\}}{1-t}$. Here, $p_{1|t}(\mathbf{s}_1|\mathbf{s}_t)$ is the denoising distribution that predicts the clean sequence given the noisy sequence, which we approximate using a neural network.

**Sampling** As shown in Algorithm 1, we initialise all sequence positions with mask tokens. Given $N$ protein backbone conformations and non-protein structure information, we employ a two-stage sampling process that integrates both sequence design and side chain packing. For each conformation, we first estimate the conditional probability $p^n(\mathbf{s}_1|\mathbf{s}_t)$ using our denoising network $f_\theta$. These individual predictions are then ensembled to obtain a more robust estimate of the sequence probability $p(\mathbf{s}_1|\mathbf{s}_t)$. To generate the next sequence state, we compute the rate matrix $R_t(\mathbf{s}_t, j)$ by taking the expectation over the conditional rate matrix with respect to our predicted $p(\mathbf{s}_1|\mathbf{s}_t)$. The next sequence state is then sampled according to a categorical distribution defined by the current state and the rate matrix.

Although many existing protein inverse folding methods do not consider side chains, side chain-ligand interactions are often crucial for designing functional proteins. Therefore, we perform side chain packing concurrently with sequence sampling (see Figure 1). For each sampled sequence $s_1$, we predict its side chain conformations $\chi$ using a dedicated side chain packing network PIPPACK (Randolph & Kuhlman, 2024) that takes into account both the protein backbone and the current sequence state. Then, for positions where $s_t$ contains mask tokens, we remove the corresponding side chain atoms to maintain consistency between the sequence and structure representations. This iterative process continues until $t$ reaches 1, at which point we obtain our final sequence $s_1$.

We also propose an adaptive sampling scheme that takes advantage of the varying uncertainty in amino acid predictions across different positions. Given a backbone structure,

**Algorithm 1** ADFLIP Sampling Process

> **Input:**
> $N$ protein and non-protein structures $\{x_1, ..., x_N\}$
> Initial sequence $s_0 = (\mathtt{m}, ..., \mathtt{m})$
> Initial side chains $\chi_0 = \emptyset$
> Time step $\Delta t$
> Denoising network $f_\theta$, side chain packing network $g_\eta$
> Initialise $t = 0$
> **while** $t < 1$ **do**
>    **for** $n = 1$ **to** $N$ **do**
>       Compute $p^n(\hat{\mathbf{s}}_1) = f_\theta(s_t, x_n, \chi_t)$
>    **end for**
>    $p(\hat{s}_1) = \frac{1}{N} \sum_n p^n(\hat{s}_1)$
>    Sample $s_1 \sim p(\hat{s}_1)$
>    **for** $n = 1$ **to** $N$ **do**
>       Compute $\chi_1^n = g(s_1, x_n)$
>    **end for**
>    Compute $R_t(\mathbf{s}_t, j) = \mathbb{E}_{p_{1|t}(\mathbf{s}_1|\mathbf{s}_t)}[R_t(\mathbf{s}_t, j|\mathbf{s}_1)]$
>    Sample $s_{t+\Delta t}$ by Equation 1
>    $t \leftarrow t + \Delta t$
> **end while**
> **Return:** Final sequence $s_1$

many of the side chains inside the core of the protein have constrained amino acid choices due to the available space. These residues display peaked categorical distributions in the denoiser's prediction. To exploit this property, we compute the purity score (maximum probability in the categorical distribution) for each masked position. As shown in Algorithm 2 (see Appendix B), positions where the purity score exceeds a threshold $\tau$, are sampled first, as these represent high-confidence predictions. After these positions are sampled, their side chain conformations are predicted and incorporated as additional structural context for subsequent steps. The time increment is then computed proportionally to the number of positions sampled. This process iterates until the remaining positions reach the purity threshold or the sampling process converges. For positions that never achieve the confidence threshold, we perform independent sampling at the final step. This adaptive scheme allows the model to progress more quickly through high-confidence positions, while expending more compute on challenging positions. Concurrent work, such as (Peng et al., 2025), adopts a similar strategy by using path planning to adaptively sample time steps, where denoiser entropy is used to determine the next step in the diffusion process.

**Training** We train a denoising network, $f_\theta(\cdot, t)$, to approximate $p_{1|t}(\mathbf{s}_1|\mathbf{s}_t, \mathbf{x}, \chi_t)$. The neural network takes as input the protein complex structure $\mathbf{x}$ and the partial side chain structure information $\chi_t$. During training, we first sample a noisy sequence $\mathbf{s}_t \sim p_{t|1}(\mathbf{s}_t|\mathbf{s}_1)$ using the closed-form conditional distribution, and we remove its current

side chains. The neural network then extracts information at both the atom and residue level to predict the probability distribution over amino acids at each position $\hat{\mathbf{s}}_1$. We optimise the network parameters using the cross-entropy loss:

$$\mathcal{L}_{\mathrm{CE}} = \mathbb{E}_{p_{\mathrm{data}}(\mathbf{s}_1)\mathcal{U}(t;0,1)p_{t|1}(\mathbf{s}_t|\mathbf{s}_1)} \left[\log p_{1|t}^\theta(\mathbf{s}_1|\mathbf{s}_t, \mathbf{x}, \chi_t)\right]$$

### 3.2. Multi-scale GNN denoiser

The GNN denoiser is a multi-scale network that aims to mix information between two different hierarchies — residues and their constituent atoms. It takes inspiration from AlphaFold3 (Abramson et al., 2024) and LigandMPNN (Dauparas et al., 2023). The GNN has two types of nodes: residue nodes and atom nodes. Each amino acid and nucleotide is represented as a residue node, while each non-hydrogen atom is represented as an atom node. Residue features have information about the arrangement of residues in the wider context of the protein backbone, whereas local information about the arrangement of atoms is contained in atom features. The network is divided into three main parts: an initial input embedding; multiple layers of processing on both the atom and residue features; and, finally, multiple decoding layers to translate the features to sequence prediction logits. Key to our approach is the information transfer between the atom and residue node types, as we alternate between the processing of local information on atom nodes and the processing of global context in residue nodes (Figure 2a).

**Atom encoder** Information about atoms is embedded in the *Atom Encoder* module, which contains Fourier embeddings (Abramson et al., 2024) and word embedding layers. The Fourier embeddings embed the sequence position of each node: atoms in their residues and residues within the overall complex. The chain identity of each node is also embedded through the chain index. The word embedding layers embed categorical variables that specify the identity of the node, such as the class of the residue, the element, and the type of the residue — whether it is a protein, a nucleotide, or an ion. These features are concatenated, projected and then processed by a module that is similar to the *Transition Block* in AlphaFold3 (Abramson et al., 2024). Furthermore, we encode the time step of the flow matching process similarly to Nichol et al. (2021).

**Residue encoder** Information about residues is encoded through the *Residue Encoder* module, which is similar to the encoder used in LigandMPNN (Dauparas et al., 2022). Briefly, each protein residue receives distance and orientation information from its nearest protein residue neighbours, together with distance information from *all* non-protein atoms. Each protein residue's $k$ nearest neighbours are calculated using the positions of their C$\alpha$ atoms. Next, dis-

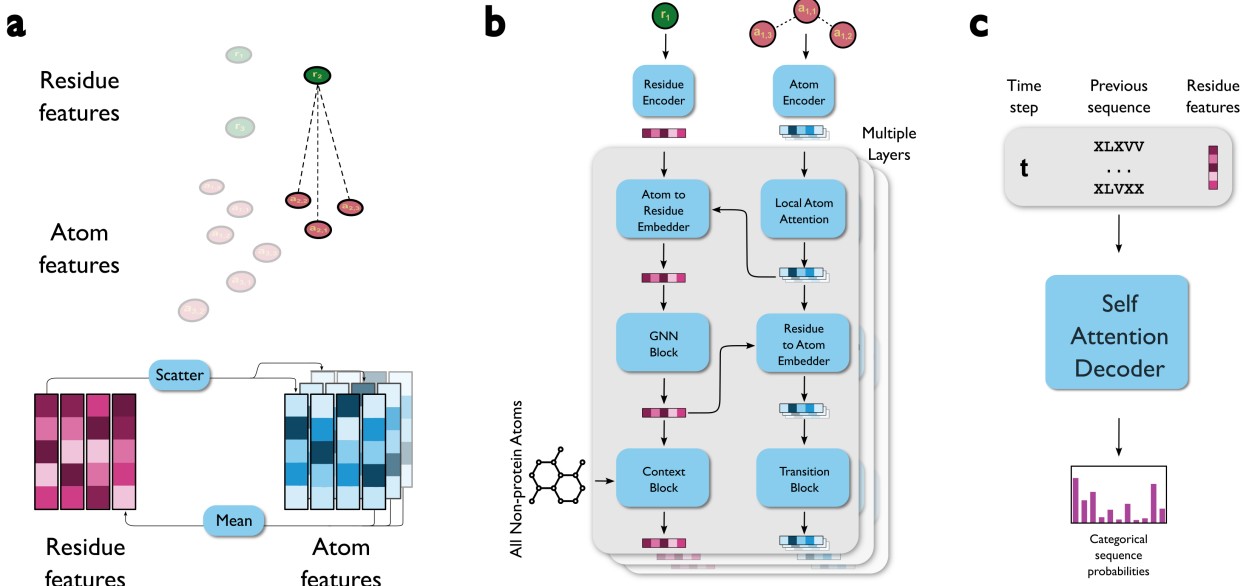

*Figure 2.* Architecture of the ADFLIP denoiser. **a**, The information flow between residue features and atom features. Updating residue features from atom features involves averaging over all atoms in a residue; updating atom features from residue features requires scattering the same residue feature over all atoms in that residue. **b**, Schematic showing the main blocks of the graph neural network (GNN) encoder. **c**, Schematic showing the inputs and output of the self-attention decoder.

tances are calculated between the backbone atoms of these protein residues and the backbone atoms of their neighbours, and the resulting distances are encoded through a radial basis function. Distances between the backbone atoms of each protein residue and *all* non-protein atoms are calculated and encoded similarly. In addition, for each non-protein atom, we also calculate a tetrahedral angle with the $C\alpha$, N, and C atoms of each protein residue. Node features for protein residues consist of a token-encoding of the residue type, along with a learned embedding of the distance features described above. Node features for the non-protein atoms include their positioning in the table of periodic elements and an additional tokenisation that encodes the identity of the atom in a one-hot manner. Edge features are projections of the distance and angle features described earlier.

**Processing trunk**    The main computation in the GNN happens in its processing trunk (Figure 2b), where information flows between the two different types of nodes and edges. Processing occurs in a hierarchical manner. Atom nodes first attend to their nearest neighbours in the *Local Atom Attention* module. This module is similar to AlphaFold3's Invariant Point Attention module, except that it also uses frame averaging (Puny et al., 2021; Huang et al., 2024) to make it more robust to rotations. The resulting contextual information from nearby atoms is then propagated to the residue features using the *Atom to Residue Embedding*. In this module, the features are first processed by a linear layer followed by a ReLU activation function (Fukushima, 1969)

and then all the atom features corresponding to a residue are averaged and then added to the residue features. The updated features are integrated in the *GNN Block*, which is a message passing layer similar to the one used in LigandMPNN. We apply diffusion-time modulation (Peebles & Xie, 2023) to the features of this message passing network, to condition the network to treat early time steps during sampling and later ones differently. The updated information from the residue nodes then feeds back to the atom nodes through the *Residue to Atom Embedder* module. This module transforms residue features using a linear layer followed by a ReLU activation. The output is then scattered to the atom features corresponding to each residue. This is followed by the *Transition Block* and the optional *Context Block*, which does message passing from the non-protein atom features to all protein features. We included this block to further increase the amount of information that flows from the non-protein context to the protein residues.

**Decoding layers**    The decoder (Figure 2c) consists of three layers of a Transformer (Vaswani et al., 2017) with a GeLU activation (Hendrycks & Gimpel, 2016). It is only applied to the residue nodes. From there, the logits are predicted using a linear layer.

### 3.3. Training-free guidance sampling

A key advantage of flow matching is its ability to incorporate external regressors to guide the generative process to-

*Table 1.* Interaction residue recovery rate on all-atom complex structure.

| Metric / Model | Ligand | Nucleotide | Metal Ion |
|---|---|---|---|
| **Perplexity ↓** | | | |
| PIFOLD | 4.03 | 7.26 | 7.55 |
| PROTEINMPNN | 4.46 | 6.48 | 4.59 |
| LIGANDMPNN | 3.84 | 4.97 | 2.73 |
| ADFLIP | **3.57** | **4.86** | **2.61** |
| **Recovery Rate (%) ↑** | | | |
| PIFOLD | 59.20 | 38.41 | 47.55 |
| PROTEINMPNN | 54.48 | 40.29 | 54.09 |
| LIGANDMPNN | 59.21 | 46.14 | 69.31 |
| ADFLIP | **62.19** | **50.21** | **75.73** |

ward samples with desired properties. Traditional regressor-guided generation requires training a regressor $p(y \mid \mathbf{s}_t)$ (Song et al., 2021; Vignac et al., 2023; Wang et al., 2024) to predict properties from intermediate states. However, retraining sophisticated neural networks to handle intermediate states $s_t$ instead of complete samples $s_1$ is often prohibitively expensive. For example, AlphaFold2's confidence scores are widely used in protein design to assess foldability and protein-protein interactions (Pacesa et al., 2024), but retraining AlphaFold2 to use masked sequences as input would be impractical.

Here, we propose a training-free approach to leverage existing regressors $p_\phi(y \mid \mathbf{s}_1)$ that only operate on complete, unmasked samples. Similar to how we define the marginal distribution $p_t(\mathbf{s}_t)$ through expectation over completions, we estimate the property predictions for intermediate states by taking the expectation of the regressor outputs $p_\phi(y \mid s_1)$ under the denoiser output $p_\theta(s_1 \mid s_t)$:

$$\hat{p}(y \mid s_t) = \mathbb{E}_{p_\theta(s_1 \mid s_t)}[p_\phi(y \mid s_1)].$$

This approach enables leveraging powerful existing regressors in a plug-and-play manner for guided generation, without requiring expensive retraining or architectural modifications.

## 4. Experiments

We evaluated ADFLIP in three scenarios of inverse folding: on all-protein complexes with small molecule ligands, nucleotides and metal ions; on ensembles of structures from dynamic complexes; and on protein-ligand complexes where we guided sequence generation towards higher predicted binding affinities.

### 4.1. All-atom inverse folding

**Test design** We evaluated ADFLIP on all-atom protein structures from the Protein Data Bank (PDB), following

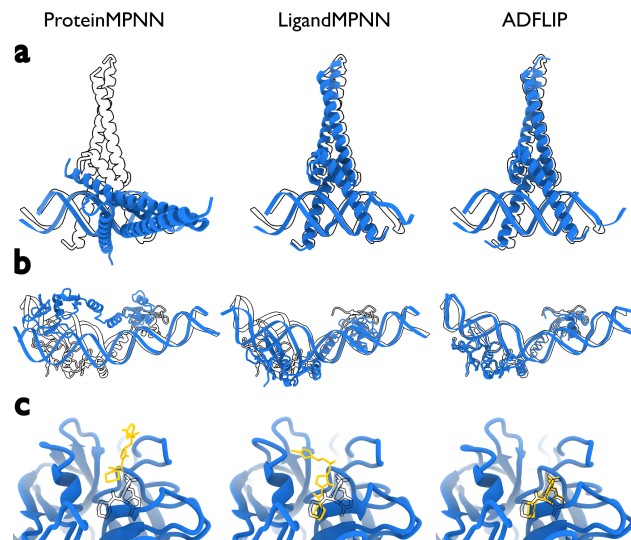

ProteinMPNN   LigandMPNN   ADFLIP

*Figure 3.* Comparison of the inverse folding of three structures with nucleotides and ligands generated with ProteinMPNN, LigandMPNN, and ADFLIP. The sequences are refolded (coloured) with Chai-1 (Chai Discovery, 2024) and compared to the starting structure (black outline). **a**, Comparison for a helix–loop–helix transcriptional activator protein (PDB ID: 1AM9, see Parraga et al., 1998), showing that ADFLIP's integration of nucleotide context information allows superior performance to ProteinMPNN. **b**, Comparison for a HNH homing endonuclease (PDB ID: 1U3E, see Shen et al., 2004) further demonstrating ADFLIP's performance for DNA-binding proteins. **c**, A close-up of the inhibitor-binding pocket of Factor Xa (PDB ID: 2UWP, see Young et al., 2007), demonstrating ADFLIP's superior performance for ligand-binding proteins. The refolded ligand (yellow) is docked closer to the target (black outline) for the ADFLIP-designed protein, but not the others.

the dataset curation protocol of LigandMPNN (Dauparas et al., 2023). Specifically, we include X-ray crystallography or cryo-EM entries that were deposited after 16 December 2022, with resolutions better than 3.5 Å, and total protein length less than 6,000 residues. We use the same validation and test sets as LigandMPNN for evaluation, comprising 317 protein complexes with small molecule ligands, 74 complexes with nucleic acids, and 83 proteins with bound metal ions. Because LigandMPNN did not release their training clusters, we cluster the remaining structures using MMseqs2 at 30% sequence identity to prevent homology between training and test sets.

We evaluated the quality of predicted protein sequences using two metrics: *perplexity* and *recovery rate*. Perplexity is calculated as the exponential of the average negative log-likelihood of the sequence. Lower perplexity indicates a better fit of the predicted amino acid probabilities and the original sequence. The recovery rate measures sequence identity between the generated sequence that is sampled

from the predicted probabilities and the original sequence. Both metrics are calculated only on residues that are close to the non-protein molecules, i.e. which have atoms within 5.0 Å of any of the non-protein atoms. For each complex structure, we sampled 10 sequences and computed the mean sequence recovery rate.

We compared our method against several baselines. Our primary comparison was with LigandMPNN (Dauparas et al., 2023)[1], which also considers the atom-level details of non-protein molecules. We retrained LigandMPNN with the same clustering we used. For a more comprehensive evaluation, we also included methods that only consider protein backbone information: PiFold (Gao et al., 2023b)[2], and ProteinMPNN (Dauparas et al., 2022)[3].

**Sequence recovery**  As shown in Table 1, methods that incorporate all-atom information of non-protein molecules significantly outperformed backbone-only approaches. For metal-binding sites, sequence recovery improved from 40% with ProteinMPNN to 69% with LigandMPNN, and further to 75% with ADFLIP. This improvement is likely due to the highly conserved nature of metal-binding amino acids. For example, in many natural metalloproteins, metal ions are coordinated by the side chains of only a few amino acids, such as histidine, cysteine, methionine, tyrosine, aspartate or glutamate (Trindler & Ward, 2017). The inclusion of explicit ligand information also improved performance for ligand-binding residues, ADFLIP achieved 62.9% recovery compared to 59.2% for LigandMPNN, 54.98% for Protein-MPNN, and 59.2% for PiFold. Similarly, for nucleotide-binding residues, ADFLIP reached 50.21% recovery, outperforming LigandMPNN (46.1%), ProteinMPNN (40.2%), and PiFold (38.1%).

**Foldability**  An important metric in inverse folding is the *foldability*, which measures whether the predicted structure for a generated sequence matches the original structure. We used Chai-1 (Chai Discovery, 2024) without MSA information to predict structures for the sequences generated by both ADFLIP and LigandMPNN. We assessed structural similarity to the reference structure from the PDB using RMSD and TM-score metrics (Mukherjee & Zhang, 2009), alongside the confidence scores from the structure prediction (pLDDT). We define a generated sequence as foldable if its predicted structure achieves a TM-score matching or exceeding 0.5 when compared to its reference structure. Our evaluation spanned all protein complexes in the test dataset with fewer than 5 ligands.

As shown in Table 2, both ADFLIP and LigandMPNN

---

[1]https://github.com/dauparas/LigandMPNN
[2]https://github.com/A4Bio/PiFold
[3]https://github.com/dauparas/ProteinMPNN

showed good performance for complexes with small molecule ligands, achieving average RMSD scores below 1.3 Å, TM-scores above 0.95, and foldability exceeding 98%. For nucleotide complexes, we observed higher RMSDs (around 5 Å), with 78-79% of the generated sequences successfully folding into structures similar to the reference. For complexes with metal ions, both methods achieve RMSDs around 2 Å and foldability scores above 80 %. Notably, both methods achieved high pLDDT scores (bigger than 95%), indicating high confidence in structure prediction. Across all three types of complexes, ADFLIP outperformed LigandMPNN, with RMSD reductions of 0.06 Å for small molecule ligands, 0.44 Å for nucleotides, and 0.1 Å for metal ions.

*Table 2.* Numerical comparison between generated sequence structure and the native structure.

| Model | RMSD (Å) | TM-score | pLDDT | Foldability |
|---|---|---|---|---|
| **Small Molecule** | | | | |
| LigandMPNN | 1.21 | 0.95 | 94.6 | 98.8% |
| ADFLIP | 1.15 | 0.96 | 90.6 | 100.0% |
| **Nucleotide** | | | | |
| LigandMPNN | 5.99 | 0.76 | 84.7 | 79.4% |
| ADFLIP | 5.55 | 0.77 | 87.5 | 83.5% |
| **Metal Ions** | | | | |
| LigandMPNN | 2.37 | 0.79 | 97.85 | 83.7% |
| ADFLIP | 2.27 | 0.78 | 98.38 | 85.1% |

### 4.2. Inverse folding on structure ensembles

**Test design**  To assess the performance of ADFLIP for inverse folding of structurally dynamic complexes, we constructed a new dataset of protein structure ensembles that were determined by solution-state NMR spectroscopy. Specifically, we selected PDB entries that were deposited after January 2015, which contained more than 100 residues, and at least two conformational states. We used MMseqs2-based clustering (Steinegger & Söding, 2017) at 30% sequence identity to remove redundant ensembles. Our final curated dataset comprised 219 ensembles, with an average of 18 conformational states per structure.

We again assessed performance using perplexity and sequence recovery rate, comparing the use of a single structure from the ensemble with the use of the entire ensemble for sequence generation. For sequence generation with a single structure, we selected the structure that exhibits the most extensive interactions with the non-protein elements from the ensemble.

**Sequence recovery**  As shown in Table 3, incorporating multiple conformational states improved sequence recovery rates: 8.6% for small molecule interactions and 5.8% for

Table 3. Recovery rate performance of NMR dataset.

| Metric / Model | Ligand | Nucleotide | Metal Ion |
|---|---|---|---|
| **Perplexity ↓** | | | |
| LigandMPNN | 7.82 | 9.37 | 3.27 |
| ADFLIP(Single) | 7.34 | 8.71 | 3.12 |
| ADFLIP(Multiple) | **5.58** | **6.91** | **2.69** |
| **Recovery Rate (%) ↑** | | | |
| LigandMPNN | 40.58 | 32.63 | 64.82 |
| ADFLIP(Single) | 41.48 | 34.48 | 65.44 |
| ADFLIP(Multiple) | **50.08** | **40.33** | **68.01** |

Table 4. Numerical comparison between generated sequence structure and the native structure for NMR dataset. Single indicates sequence generation using an individual conformational state, while Multiple indicates sequence generation using the complete ensemble of NMR conformational states.

| Model | RMSD (Å) | TM-score | pLDDT | Foldability |
|---|---|---|---|---|
| LigandMPNN | 8.34 | 0.68 | 78.4 | 84.5% |
| ADFLIP (Single) | 9.10 | 0.67 | 79.5 | 81.7% |
| ADFLIP (Multiple) | 7.21 | 0.69 | 80.19 | 82.5% |

nucleotide interactions. These improvements were achieved using the same model, without fine-tuning, demonstrating ADFLIP's inherent ability to integrate information from multiple structural states during sequence generation. For metal ion interactions, we observed a more modest improvement of 2.6% in sequence recovery. This is likely because metal-binding sites in proteins typically maintain relatively stable conformations, even in the dynamic protein ensembles captured by NMR (which was also reflected in the higher overall recovery rates for metal-binding proteins in the NMR test set).

**Foldability** We assessed the foldability of the generated sequences by predicting their structures using Chai-1. We compared the predicted structure with the structure from the original NMR ensemble that had the lowest RMSD with the predicted structure. Although generating sequences for dynamic protein complexes remains difficult (as evidenced by high RMSDs, low TM-scores, low pLDDT scores and low foldabilities), using ensemble information in ADFLIP reduces the average RMSD from 9.11 to 7.61 Å compared to using a single structure (see Table 4).

### 4.3. Guidance by binding affinity

We assessed guidance-based sequence generation in AD-FLIP by incorporating DSMBind (Jin et al., 2024) predictions of ligand binding affinity. DSMBind is an unsupervised, deep learning-based predictor of binding affinity that takes as input the complex structure (including side chain information), protein sequences, and ligand information. By

Table 5. Ligand binding affinity guided results

| Model | Affinity gain | Foldability |
|---|---|---|
| ADFLIP (Unguided) | 41.9% | 100% |
| ADFLIP (Guided) | 58.1% | 91.4% |

incorporating DSMBind predictions in the flow matching process, we aimed to guide the generation process towards sequences that are predicted to have higher binding affinity for their target ligands.

We tested our guidance-based approach using the subset of 210 protein-ligand complexes from the test set described in Section 4.1 that contain a single ligand. First, we used DSMBind to establish baseline binding affinities for the wild-type sequences. We then set our generation target to achieve a 10% improvement over the wild-type binding affinities. For each structure, we generated 10 different sequences and evaluated their predicted binding affinities using DSMBind. To assess the effectiveness of our guidance approach, we compared guided and unguided sequences generated by ADFLIP.

We evaluated the effectiveness of our approach using two metrics: (1) affinity gain — whether the generated sequence's predicted binding affinity exceeds that of the wild-type sequence and (2) foldability — whether the generated sequence's predicted structure maintains structural similarity to the input backbone, i.e. with TM-score $\geq 0.5$.

As shown in Table 5, our guidance-based approach significantly improves the success rate of generating sequences with enhanced binding affinity from 41.9% to 58.1%, while maintaining foldability above 90%. However, we note that by using a fixed protein backbone structure, and limited side chain sampling, the ability of DSMBind to achieve better binding affinity is currently limited. Refinement of the backbone structure in our method may yield further improvements in the future.

## 5. Discussion

The ultimate goal of protein design is to create proteins with improved or new functionalities compared to those existing in nature. Because protein functionality is typically coupled to conformational change, and it often involves a wide range of chemistry, the design of new protein functionality will need to consider both all-atom structures and the requirement of a single sequence to adopt multiple conformations. ADFLIP represents a step in this direction. Its all-atom model improves sequence recovery rates compared to current state-of-the-art for protein complexes with ligands, nucleotides, or metal ions. Leveraging the flexibility of discrete flow matching, ADFLIP also allows incorpo-

rating multiple conformational states in the sequence generation process, as demonstrated for structure ensembles determined by NMR. Moreover, the flow matching process allows steering sequence generation towards desired properties through training-free guidance sampling, as demonstrated by the design of higher affinity ligand binders using DSMBIND (Jin et al., 2024) as a plug-in regressor.

Although ADFLIP shows strong performance in sequence recovery for complexes with small molecule ligands, and particularly with metal ions, lower recovery rates for complexes with nucleotides highlight opportunities for future advances. Work by Joshi et al. (2024) and Nori & Jin (2024) demonstrates that incorporating conformational dynamics benefits the design of RNA structures. It is possible that the inherently increased conformational dynamics of nucleotide complexes is not captured sufficiently by a single conformational state. Whereas we show that ADFLIP can condition on multiple conformational states, it is not yet clear how one would best obtain a range of useful conformational states, for protein-nucleotide complexes specifically, or any biologically active complex in general.

One future direction is to integrate structural refinement into the sequence generation process, similar to the RNAflow (Nori & Jin, 2024) approach in RosettaFold2NA. As mentioned in section 4.3, performing structure refinement during the sampling of sequences will also be beneficial for guiding-based sampling. Future versions of ADFLIP could therefore incorporate backbone flexibility to allow for a more comprehensive exploration of protein sequence-structure relationships, and hence the design of better protein functionality.

## Acknowledgements

We are grateful to Xiongwen Ke for helpful discussion and Bogdan Toader for critical reading of the paper. We also thank Jake Grimmett, Toby Darling and Ivan Clayson for help with high-performance computing. This work was supported by the Medical Research Council, as part of the UK Research and Information (MC_UP_A025_1013 to S.H.W.S.).

## Impact Statement

This paper contributes to the field of all-atom protein design by enabling sequence generation for complex and dynamic biomolecular assemblies. Our work advances generative modelling for molecular design, with potential applications in synthetic biology, and biotechnology. While ADFLIP-designed sequences have not yet been experimentally validated, such validation is essential for any real-world application.

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

## A. Dataset

We follow the dataset generation process used in LigandMPNN, collecting protein assemblies from the Protein Data Bank (PDB) as of December 16, 2022, that were determined by X-ray crystallography or cryo-electron microscopy at a resolution better than 3.5 Å and contain fewer than 6,000 residues. Residues introduced solely to aid crystallisation (e.g., ZMR, Z9D) were removed. Following the AlphaFold3 paper, such residues are parsed as glycans. A full list of excluded crystallisation-aid residues and glycan residues is provided in Appendix D.

We cluster each protein chain at a 30% sequence identity threshold using MMseqs2 (Steinegger & Söding, 2017). We hold out a non-overlapping subset of clusters corresponding to distinct interaction contexts: 317 protein–small molecule complexes, 74 protein–nucleotide complexes, and 83 protein–metal ion complexes. The full list of held-out PDB entries is available in Appendix D. The resulting training dataset comprises 27,818 clusters. For both training and evaluation, we use the same PDB collection as in the LigandMPNN repository[4].

## B. Purity Sampling

The following algorithm describes the adaptive time step sampling strategy used in ADFLIP. Starting from a fully masked sequence, the method employs a denoising network $f_\theta$ to predict amino acid distributions. Unlike fixed-timestep approaches, this method adaptively selects high-confidence positions for sampling based on a *purity threshold* $\tau$. Only positions exceeding this threshold are updated at each step, allowing for progressive and focused refinement. A side chain packing network $g_\eta$ is then applied to model side chains at the newly sampled positions. The process iterates until all positions are resolved, the model's confidence saturates, or the maximum number of iterations $K$ is reached. If unresolved positions remain at the end, they are sampled in a final fallback step.

---

**Algorithm 2** ADFLIP Adaptive Sampling Process

---

**Input:**
 $N$ protein backbone conformation and non-protein structures $x_1, ..., x_N$
 Initial sequence $s_0 = (\text{m}, ..., \text{m})$
 Initial side chains $\chi_0 = \emptyset$
 Purity threshold $\tau$, maximum iterations $K$
 Denoising network $f_\theta$, side chain packing network $g_\eta$
 Initialise $t = 0$, $M =$ positions still masked, $k = 0$
 **while** $M > 0$ **and** $t < 1$ **and** $k < K$ **do**
   **for** $n = 1$ **to** $N$ **do**
     Compute $p^n(\hat{s}_1) = f_\theta(s_t, x_n, \chi_t)$
   **end for**
   $p(\hat{s}_1) = \frac{1}{N} \sum_n p^n(\hat{s}_1)$
   Compute purity scores $\phi_i = \max_j p(\hat{s}_1^{(i)} = j)$ for masked positions
   $I = i : \phi_i > \tau$ {Positions to sample}
   Sample $s_1^{(i)} \sim p(\hat{s}_1^{(i)})$ for $i \in I$
   **for** $n = 1$ **to** $N$ **do**
     Compute $\chi_1^n = g(s_1, x_n)$ for positions in $I$
   **end for**
   $M \leftarrow M - |I|$ {Update mask count}
   $t \leftarrow t + |I|/L$ {$L$ is sequence length}
   $k \leftarrow k + 1$
 **end while**
 **if** $k = K$ **and** $M > 0$ **then**
   Let $J = j : s_t^{(j)} = \text{m}$ {Remaining masked positions}
   Sample $s_1^{(j)} \sim p(\hat{s}_1^{(j)})$ for $j \in J$
 **end if**
 **Return:** Final sequence $s_1$

---

[4] https://github.com/dauparas/LigandMPNN/

## C. Training-free classifier Guidance Sampling

The following algorithm outlines the training-free classifier guidance sampling strategy used in ADFLIP. Starting from a fully masked sequence, the method uses a denoising network $f_\theta$ to produce amino acid distributions $p(\hat{s}_1)$ conditioned on the input structures. Unlike standard sampling-based approaches, we do not sample a sequence from this distribution. Instead, we directly feed the probability distribution $p(\hat{s}_1)$ into a regressor network $h_\phi$ to predict the target property value $\hat{y}$. The guidance signal is derived from the discrepancy between the predicted value and the desired target $y$, approximated as a pseudo-gradient $-\nabla_{s_1}\|y - \hat{y}\|^2$, which is used to reweight the sequence distribution. This guides the sampling process toward sequences more likely to yield the desired property. A side chain packing network $g_\eta$ assigns side chains for each structure. Reverse-time sampling with a fixed time step $\Delta t$ is performed iteratively (e.g., using Eq. 1) until the full sequence is generated.

---

**Algorithm 3** Training-free classifier Guidance Sampling Process

---

**Input:**
$N$ protein and non-protein structures $\{x_1, ..., x_N\}$
Initial sequence $s_0 = (\mathtt{m}, ..., \mathtt{m})$
Initial side chains $\chi_0 = \emptyset$
Time step $\Delta t$
Denoising network $f_\theta$, side chain packing network $g_\eta$, regressor network $h_\phi$
Target value $y$
Initialise $t = 0$
**while** $t < 1$ **do**
    **for** $n = 1$ **to** $N$ **do**
        Compute $p^n(\hat{\mathbf{s}}_1) = f_\theta(s_t, x_n, \chi_t)$
    **end for**
    $p(\hat{s}_1) = \frac{1}{N}\sum_n p^n(\hat{s}_1)$
    Compute $\hat{y}$ by $p(\hat{s}_1)$: $\hat{y} = h_\phi(p(\hat{s}_1))$
    Approximate $p(y|s_1) \approx -\nabla_{s_1}\|y - \hat{y}\|^2$
    Sample $s_1 \sim p(\hat{\mathbf{s}}_1)p(y|\hat{s}_1)$
    **for** $n = 1$ **to** $N$ **do**
        Compute $\chi_1^n = g(s_1, x_n)$
    **end for**
    Compute $R_t(\mathbf{s}_t, j) = \mathbb{E}_{p_{1|t}(\mathbf{s}_1|\mathbf{s}_t)}[R_t(\mathbf{s}_t, j|\mathbf{s}_1)]$
    Sample $s_{t+\Delta t}$ by Eq 1
    $t \leftarrow t + \Delta t$
**end while**
**Return:** Final sequence $s_1$

---

## D. PDB IDs and CCD codes

**PDB IDs for the small molecule interaction benchmark** 1A28, 1BZC, 1DRV, 1E3G, 1ELB, 1ELC, 1EPO, 1F0R, 1G7F, 1G7G, 1GVW, 1GX8, 1I37, 1KAV, 1KDK, 1KV1, 1L8G, 1LHU, 1LPG, 1NC1, 1NFX, 1NHZ, 1NL9, 1NNY, 1NWL, 1ONY, 1PYN, 1QB1, 1QKT, 1QXK, 1R0P, 1SJ0, 1SQN, 1V2N, 1XJD, 1XWS, 1YC1, 1YQJ, 1Z95, 1ZP8, 2AYR, 2B07, 2B4L, 2BAJ, 2BAK, 2BAL, 2BSM, 2CET, 2E2R, 2F6T, 2FDP, 2G94, 2HAH, 2IHQ, 2IWX, 2J2U, 2J34, 2J4I, 2J94, 2J95, 2O0U, 2OAX, 2OJG, 2OJJ, 2P4J, 2P7G, 2P7Z, 2POG, 2QBP, 2QBQ, 2QBS, 2QE4, 2QMG, 2UWL, 2UWO, 2UWP, 2V7A, 2VH0, 2VH6, 2VKM, 2VRJ, 2VW5, 2VWC, 2W8Y, 2WC3, 2WEB, 2WEC, 2WEQ, 2WGJ, 2WUF, 2WYG, 2WYJ, 2XAB, 2XB8, 2XDA, 2XHT, 2XJ1, 2XJ2, 2XJG, 2XJX, 2Y7X, 2Y7Z, 2Y80, 2Y81, 2Y82, 2YDW, 2YEK, 2YEL, 2YFE, 2YFX, 2YGE, 2YGF, 2YI0, 2YI7, 2YIX, 2ZMM, 3ACW, 3ACX, 3B5R, 3B65, 3BGQ, 3BGZ, 3CKP, 3COW, 3COY, 3COZ, 3D7Z, 3D83, 3EAX, 3EKR, 3FV1, 3FV2, 3FVK, 3GBA, 3GBB, 3GCS, 3GCU, 3GY3, 3HEK, 3I25, 3IOC, 3IPH, 3IW6, 3K97, 3LPI, 3LPK, 3LXK, 3M35, 3MYG, 3N76, 3NQ3, 3NYX, 3O5X, 3O8P, 3PWW, 3ROC, 3TFN, 3U81, 3UEU, 3UEV, 3UEW, 3UEX, 3VHA, 3VHC, 3VHD, 3VJE, 3VVY, 3VW1, 3VW2, 3WHA, 3WZ6, 3WZ8, 3ZC5, 3ZM9, 3ZZE, 4A4V, 4A4W, 4A7I, 4AG8, 4AP7, 4B6O, 4B9K, 4CD0, 4CGA, 4CMO, 4DA5, 4E5W, 4E6D, 4E9U, 4EA2, 4EGK, 4ER1, 4FCQ, 4FFS, 4FLP, 4G8N, 4GNY, 4GU6, 4HGE, 4IGT, 4K0Y, 4K9Y, 4KAO, 4KCX, 4LYW, 4M0R, 4M12, 4M13, 4MUF, 4NH8, 4NWC, 4OO4, 4OO5, 4OO7, 4OO9, 4OOB, 4P5Z, 4PMM, 4POP, 4QEV, 4QEW, 4QYY, 4RFM, 4RWJ, 4TWP, 4UYF, 4V01, 4W9F, 4W9L, 4WA9, 4WKN, 4X6P, 4XIP, 4XIR, 4Y79, 4YBK, 4YMB, 4YML, 4YNB, 4YTH, 4Z0K, 4ZAE, 5AA9, 5ACY, 5D26, 5D3H, 5D3J, 5D3L, 5D3T, 5DLX, 5DQC, 5DWR, 5E74, 5EGM, 5ENG, 5EQP, 5EQY, 5ER1, 5EXM, 5EXN, 5F9B, 5FTO, 5FUT, 5HCV, 5I3V, 5I3Y, 5I9X, 5I9Z, 5IE1, 5IH9, 5JQ5, 5KZ0, 5L2S, 5LLI, 5LNY, 5LSG, 5NEB, 5NW1, 5NYH, 5OP5, 5OQ8, 5QQP, 5T19, 5TPX, 5V82, 5YFS, 5YFT, 6C2R, 6CJR, 6CPW, 6DGQ, 6DGR, 6DYU, 6DYV, 6EL5, 6ELO, 6ELP, 6EY9, 6EYB, 6F1N, 6GE7, 6GF9, 6GFS, 6GHH, 6I61, 6I64, 6I67, 6MD0, 6MH1, 6MH7, 6N7A, 6N8X, 6NO9, 6NV7, 6NV9, 6OLX, 6QI7

**PDB IDs for the nucleotide interaction benchmark** 1A0A, 1AM9, 1AN4, 1B01, 1BC7, 1BC8, 1DI2, 1EC6, 1HLO, 1HLV, 1I3J, 1PVI, 1QUM, 1SFU, 1U3E, 1XPX, 1YO5, 1ZX4, 2C5R, 2C62, 2NQ9, 2O4A, 2P5L, 2XDB, 2YPB, 2ZHG, 2ZIO, 3ADL, 3BSU, 3FC3, 3G73, 3GNA, 3GX4, 3LSR, 3MJ0, 3MVA, 3N7Q, 3OLT, 3VOK, 3VWB, 3ZP5, 4ATO, 4BHM, 4BQA, 4E0P, 4NID, 4WAL, 5CM3, 5HAW, 5MHT, 5VC9, 5W9S, 5YBD, 6BJV, 6DNW, 6FQR, 6GDR, 6KBS, 6LFF, 6LMJ, 6OD4, 6WDZ, 6X70, 6Y93, 7BCA, 7C0G, 7EL3, 7JSA, 7JU3, 7KII, 7KIJ, 7MTL, 7Z0U, 8DWM

**PDB IDs for the metal interaction benchmark** 1DWH, 1E4M, 1E6S, 1E72, 1F35, 1FEE, 1JOB, 1LQK, 1M5E, 1M5F, 1MOJ, 1MXY, 1MXZ, 1MY1, 1NKI, 1QUM, 1SGF, 1T31, 1U3E, 2BDH, 2BX2, 2CFV, 2E6C, 2NQ9, 2NQJ, 2NZ6, 2OU7, 2VXX, 2ZWN, 3BVX, 3CV5, 3F4V, 3F5L, 3FGG, 3HG9, 3HKN, 3HKT, 3I9Z, 3K7R, 3L24, 3L7T, 3M7P, 3MI9, 3O1U, 3U92, 3U93, 3U94, 3WON, 4AOJ, 4DY1, 4HZT, 4I0F, 4I0J, 4I0Z, 4I11, 4I12, 4JD1, 4NAZ, 4WD8, 4X68, 5F55, 5F56, 5FGS, 5HEZ, 5I4J, 5L70, 5VDE, 6A4X, 6BUU, 6CYT, 6IV2, 6LKP, 6LRD, 6WDZ, 6X75, 7DNR, 7E34, 7KII, 7N7G, 7S7L, 7S7M, 7W5E, 7WB2

**PDB IDs for the NMR benchmark**  8VOH, 8WLS, 8X8T, 8XZ2, 9ATN, 9C5E, 9GAG, 5K5G, 5L85, 5LSD, 5M1G, 5M8I, 5M9D, 5MPG, 5MPL, 5N8M, 5NF8, 5NKO, 5NOC, 5NWM, 5OEO, 5OGU, 5OR5, 5OWI, 5TM0, 5TMX, 5TN2, 5U4K, 5U5S, 5U9B, 5US5, 5VTO, 5W4S, 5WYO, 5X29, 5X3Z, 5XV8, 5XZK, 5YAM, 2NBV, 5ZAU, 5ZUX, 6B1G, 6B7G, 6BA6, 6WA1, 6WLH, 6XFL, 6XOR, 6Y8V, 6YP5, 6ZDB, 7A0O, 7ACT, 7B2B, 7CLV, 7CSQ, 7DEE, 7DFE, 7JQ8, 7K3S, 7K7F, 6CTB, 7LOI, 7MLA, 7ND1, 7OVC, 7PJ1, 7PKU, 7Q4L, 7QDE, 7QRI, 7QS6, 7QUU, 7RNO, 7RPM, 2MY2, 2MYJ, 2MZ1, 2MZC, 2MZD, 2MZP, 2N01, 2N0S, 2N0Y, 2N18, 2N1A, 2N1D, 2N1G, 2N1T, 2N2A, 2N2H, 2N2J, 2N3J, 2N3O, 2NBW, 2N54, 2N55, 2N5E, 2N5G, 2N64, 2N73, 2N74, 2N8A, 2N9B, 2N9P, 2N9X, 2NAO, 7S5J, 7SFT, 7T2F, 7VBG, 7VRL, 7VU7, 7WJ0, 7X5C, 7YWQ, 7ZAP, 7ZE0, 7ZEY, 7ZEX, 8BA1, 8CA0, 8COO, 8DPX, 8DSB, 8DSX, 8FG1, 8HEP, 8HEQ, 8HER, 8HPB, 8J4I, 8K2R, 8K2T, 8K31, 8ONU, 8OX2, 8PEK, 8PXX, 8R1X, 8RAJ, 8S8O, 8SG2, 8TT7, 8U9O, 2NB1, 2NBJ, 2NDG, 2NDP, 2RVN, 5B7J, 5GWM, 5H7P, 5I8N, 5IAY, 5ID3, 5IXF, 5J6Z, 5JPW, 5JTL, 5JYN, 5JYV, 6C44, 6CLZ, 6DMP, 6DSL, 6E5N, 6E8W, 6EVI, 6F0Y, 6FDT, 6G04, 6G8O, 6GBE, 6GBM, 6GC3, 6GVU, 6H8C, 6HPJ, 6IVU, 6IWJ, 6JXX, 6K3K, 6L8V, 6LMR, 6LQZ, 6LUL, 6LXG, 6M78, 6N2M, 6NHW, 6O0I, 6OQJ, 6OSW, 6Q2Z, 6QTC, 6R5G, 6RH6, 6S3W, 6SAI, 6SDW, 6SDY, 6SNJ, 6SO9, 6TDM, 6TDN, 6TL0, 6TV5, 6TVM, 6TWR, 6U19, 6U4N, 6U6P, 6U6S, 6UHW, 6UJV, 6UT2, 6V88

**Crystallisation aids**  SO4, GOL, EDO, PO4, ACT, PEG, DMS, TRS, PGE, PG4, FMT, EPE, MPD, MES, CD, IOD

**Other ligands excluded**  144, 15P, 1PE, 2F2, 2JC, 3HR, 3SY, 7N5, 7PE, 9JE, AAE, ABA, ACE, ACN, ACT, ACY, AZI, BAM, BCN, BCT, BDN, BEN, BME, BO3, BTB, BTC, BU1, C8E, CAD, CAQ, CBM, CCN, CIT, CL, CLR, CM, CMO, CO3, CPT, CXS, D10, DEP, DIO, DMS, DN, DOD, DOX, EDO, EEE, EGL, EOH, EOX, EPE, ETF, FCY, FJO, FLC, FMT, FW5, GOL, GSH, GTT, GYF, HED, IHP, IHS, IMD, IOD, IPA, IPH, LDA, MB3, MEG, MES, MLA, MLI, MOH, MPD, MRD, MSE, MYR, N, NA, NH2, NH4, NHE, NO3, O4B, OHE, OLA, OLC, OMB, OME, OXA, P6G, PE3, PE4, PEG, PEO, PEP, PG0, PG4, PGE, PGR, PLM, PO4, POL, POP, PVO, SAR, SCN, SEO, SEP, SIN, SO4, SPD, SPM, SR, STE, STO, STU, TAR, TBU, TME, TPO, TRS, UNK, UNL, UNX, UPL, URE

**CCD codes defining glycans**  045, 05L, 07E, 07Y, 08U, 09X, 0BD, 0H0, 0HX, 0LP, 0MK, 0NZ, 0UB, 0V4, 0WK, 0XY, 0YT, 10M, 12E, 145, 147, 149, 14T, 15L, 16F, 16G, 16O, 17T, 18D, 18O, 1CF, 1FT, 1GL, 1GN, 1LL, 1S3, 1S4, 1SD, 1X4, 20S, 20X, 22O, 22S, 23V, 24S, 25E, 26O, 27C, 289, 291, 293, 2DG, 2DR, 2F8, 2FG, 2FL, 2GL, 2GS, 2H5, 2HA, 2M4, 2M5, 2M8, 2OS, 2WP, 2WS, 32O, 34V, 38J, 3BU, 3DO, 3DY, 3FM, 3GR, 3HD, 3J3, 3J4, 3LJ, 3LR, 3MG, 3MK, 3R3, 3S6, 3SA, 3YW, 40J, 42D, 445, 44S, 46D, 46Z, 475, 48Z, 491, 49A, 49S, 49T, 49V, 4AM, 4CQ, 4GC, 4GL, 4GP, 4JA, 4N2, 4NN, 4QY, 4R1, 4RS, 4SG, 4UZ, 4V5, 50A, 51N, 56N, 57S, 5GF, 5GO, 5II, 5KQ, 5KS, 5KT, 5KV, 5L3, 5LS, 5LT, 5MM, 5N6, 5QP, 5SP, 5TH, 5TJ, 5TK, 5TM, 61J, 62I, 64K, 66O, 6BG, 6C2, 6DM, 6GB, 6GP, 6GR, 6K3, 6KH, 6KL, 6KS, 6KU, 6KW, 6LA, 6LS, 6LW, 6MJ, 6MN, 6PZ, 6S2, 6UD, 6YR, 6ZC, 73E, 79J, 7CV, 7D1, 7GP, 7JZ, 7K2, 7K3, 7NU, 83Y, 89Y, 8B7, 8B9, 8EX, 8GA, 8GG, 8GP, 8I4, 8LR, 8OQ, 8PK, 8S0, 8YV, 95Z, 96O, 98U, 9AM, 9C1, 9CD, 9GP, 9KJ, 9MR, 9OK, 9PG, 9QG, 9S7, 9SG, 9SJ, 9SM, 9SP, 9T1, 9T7, 9VP, 9WJ, 9WN, 9WZ, 9YW, A0K, A1Q, A2G, A5C, A6P, AAL, ABD, ABE, ABF, ABL, AC1, ACR, ACX, ADA, AF1, AFD, AFO, AFP, AGL, AH2, AH8, AHG, AHM, AHR, AIG, ALL, ALX, AMG, AMN, AMU, AMV, ANA, AOG, AQA, ARA, ARB, ARI, ARW, ASC, ASG, ASO, AXP, AXR, AY9, AZC, B0D, B16, B1H, B1N, B2G, B4G, B6D, B7G, B8D, B9D, BBK, BBV, BCD, BDF, BDG, BDP, BDR, BEM, BFN, BG6, BG8, BGC, BGL, BGN, BGP, BGS, BHG, BM3, BM7, BMA, BMX, BND, BNG, BNX, BO1, BOG, BQY, BS7, BTG, BTU, BW3, BWG, BXF, BXP, BXX, BXY, BZD, C3B, C3G, C3X, C4B, C4W, C5X, CBF, CBI, CBK, CDR, CE5, CE6, CE8, CEG, CEZ, CGF, CJB, CKB, CKP, CNP, CR1, CR6, CRA, CT3, CTO, CTR, CTT, D1M, D5E, D6G, DAF, DAG, DAN, DDA, DDL, DEG, DEL, DFR, DFX, DG0, DGO, DGS, DGU, DJB, DJE, DK4, DKX, DKZ, DL6, DLD, DLF, DLG, DNO, DO8, DOM, DPC, DQR, DR2, DR3, DR5, DRI, DSR, DT6, DVC, DYM, E3M, E5G, EAG, EBG, EBQ, EEN, EEQ, EGA, EMP, EMZ, EPG, EQP, EQV, ERE, ERI, ETT, EUS, F1P, F1X, F55, F58, F6P, F8X, FBP, FCA, FCB, FCT, FDP, FDQ, FFC, FFX, FIF, FK9, FKD, FMF, FMO, FNG, FNY, FRU, FSA, FSI, FSM, FSW, FUB, FUC, FUD, FUF, FUL, FUY, FVQ, FX1, FYJ, G0S, G16, G1P, G20, G28, G2F, G3F, G3I, G4D, G4S, G6D, G6P, G6S, G7P, G8Z, GAA, GAC, GAD, GAF, GAL, GAT, GBH, GC1, GC4, GC9, GCB, GCD, GCN, GCO, GCS, GCT, GCU, GCV, GCW, GDA, GDL, GE1, GE3, GFP, GIV, GL0, GL1, GL2, GL4, GL5, GL6, GL7, GL9, GLA, GLC, GLD, GLF, GLG, GLO, GLP, GLS, GLT, GM0, GMB, GMH, GMT, GMZ, GN1, GN4, GNS, GNX, GP0, GP1, GP4, GPH, GPK, GPM, GPO, GPQ, GPU, GPV, GPW, GQ1, GRF, GRX, GS1, GS9, GTK, GTM, GTR, GU0, GU1, GU2, GU3, GU4, GU5, GU6, GU8, GU9, GUF, GUL, GUP, GUZ, GXL, GXV, GYE, GYG, GYP, GYU, GYV, GZL, H1M, H1S, H2P, H3S, H53, H6Q, H6Z, HBZ, HD4, HNV, HNW, HSG, HSH, HSJ, HSQ, HSX, HSY, HTG, HTM, HVC, IAB, IDC, IDF, IDG, IDR, IDS, IDU, IDX, IDY, IEM, IN1, IPT, ISD, ISL, ISX, IXD, J5B, JFZ, JHM, JLT, JRV, JSV, JV4, JVA, JVS, JZR, K5B, K99, KBA, KBG, KD5, KDA, KDB, KDD, KDE, KDF, KDM, KDN, KDO, KDR, KFN, KG1, KGM, KHP, KME, KO1, KO2, KOT, KTU, L0W, L1L, L6S, L6T, LAG, LAH, LAI, LAK, LAO, LAT, LB2, LBS, LBT, LCN, LDY, LEC, LER, LFC, LFR, LGC, LGU, LKA, LKS, LM2, LMO, LNV, LOG, LOX, LRH, LTG, LVO, LVZ, LXB, LXC, LXZ, LZ0, M1F, M1P, M2F, M3M, M3N, M55, M6D, M6P, M7B, M7P, M8C, MA1, MA2, MA3, MA8, MAB, MAF, MAG, MAL, MAN, MAT, MAV, MAW, MBE, MBF, MBG, MCU, MDA, MDP, MFB, MFU, MG5, MGC, MGL, MGS, MJJ, MLB, MLR, MMA, MN0, MNA, MQG, MQT, MRH, MRP, MSX, MTT, MUB, MUR, MVP, MXY, MXZ, MYG, N1L, N3U, N9S, NA1, NAA, NAG, NBG, NBX, NBY, NDG, NFG, NG1, NG6, NGA, NGC, NGE, NGK, NGR, NGS, NGY, NGZ, NHF, NLC, NM6, NM9, NNG, NPF, NSQ, NT1, NTF, NTO, NTP, NXD, NYT, OAK, OI7, OPM, OSU, OTG, OTN, OTU, OX2, P53, P6P, P8E, PA1, PAV, PDX, PH5, PKM, PNA, PNG, PNJ, PNW, PPC, PRP, PSG, PSV, PTQ, PUF, PZU, QDK, QIF, QKH, QPS, QV4, R1P, R1X, R2B, R2G, RAE, RAF, RAM, RAO, RB5, RBL, RCD, RER, RF5, RG1, RGG, RHA, RHC, RI2, RIB, RIP, RM4, RP3, RP5, RP6, RR7, RRJ, RRY, RST, RTG, RTV, RUG, RUU, RV7, RVG, RVM, RWI, RY7, RZM, S7P, S81, SA0, SCG, SCR, SDY, SEJ, SF6, SF9, SFU, SG4, SG5, SG6, SG7, SGA, SGC, SGD, SGN, SHB, SHD, SHG, SIA, SID, SIO, SIZ, SLB, SLM, SLT, SMD, SN5, SNG, SOE, SOG, SOL, SOR, SR1, SSG, SSH, STW, STZ, SUC, SUP, SUS, SWE, SZZ, T68, T6D, T6P, T6T, TA6, TAG, TCB, TDG, TEU, TF0, TFU, TGA, TGK, TGR, TGY, TH1, TM5, TM6, TMR, TMX, TNX, TOA, TOC, TQY, TRE, TRV, TS8, TT7, TTV, TU4, TUG, TUJ, TUP, TUR, TVD, TVG, TVM, TVS, TVV, TVY, TW7, TWA, TWD, TWG, TWJ, TWY, TXB, TYV, U1Y, U2A, U2D, U63, U8V, U97, U9A, U9D, U9G, U9J, U9M, UAP, UBH, UBO, UDC, UEA, V3M, V3P, V71, VG1, VJ1, VJ4, VKN, VTB, W9T, WIA, WOO, WUN, WZ1, WZ2, X0X, X1P, X1X, X2F, X2Y, X34, X6X, X6Y, XDX, XGP, XIL, XKJ, XLF, XLS, XMM, XS2, XXM, XXR, XXX, XYF, XYL, XYP, XYS, XYT, XYZ, YDR, YIO, YJM, YKR, YO5, YX0, YX1, YYB, YYH, YYJ, YYK, YYM, YYQ, YZ0, Z0F, Z15, Z16, Z2D, Z2T, Z3K, Z3L, Z3Q, Z3U, Z4K, Z4R, Z4S, Z4U, Z4V, Z4W, Z4Y, Z57, Z5J, Z5L, Z61, Z6H, Z6J, Z6W, Z8H, Z8T, Z9D, Z9E, Z9H, Z9K, Z9L, Z9M, Z9N, Z9W, ZB0, ZB1, ZB2, ZB3, ZCD, ZCZ, ZD0, ZDC, ZDO, ZEE, ZEL, ZGE, ZMR

