# OpenReview forum: "All-atom inverse protein folding through discrete flow matching"
_ICML.cc/2025/Conference — ICML 2025 poster_

### Official Review · Reviewer_dLqF · 2025-03-12

**Overall Recommendation:** 4

**Summary:**

All-atom Discrete Flow Matching Inverse Protein Folding (ADFLIP) is a generative model for designing protein sequences conditioned on full atomic structures. Unlike existing inverse folding methods, ADFLIP progressively incorporates predicted side chains during sequence generation. Additionally, ADFLIP employs training-free classifier guidance to optimize sequences using pre-trained models. Evaluations on protein-ligand, nucleotide, and metal ion complexes, show that ADFLIP achieves state-of-the-art performance in both single-structure and multi-structure inverse folding tasks, highlighting its potential for all-atom protein design.

**Claims And Evidence:**

As a primary comparison to LigandMPNN, ADFLIP demonstrates comprehensive evidence of its claims.

**Essential References Not Discussed:**

Concurrent work so not expected to compare but FAMPNN [1] is worthwhile to discuss.

[1] https://www.biorxiv.org/content/10.1101/2025.02.13.637498v1.full.pdf

**Experimental Designs Or Analyses:**

Yes non guided experiements are sound but could benefit from deeper analysis given the small test set.

**Methods And Evaluation Criteria:**

Inverse folding benchmarks and evaluations are clear but could use error bars. For example, in Table 1, for each complex structure, 10 sequences were sampled. Given the test set is not large it would be interesting to see the sequence recovery distribution or what is the per sample average and std delta of improvement over LigandMPNN rather than a single globular metric.

Given here ADFLIP and LigandMPNN are trained on the same data, the benchmarks are fair. It would also be important to add the results for the public LigandMPNN weights which could point to the importance of the training set filters.


The DSMBind-based guidance seems odd as it is a structure-based method. As a result, you generate nondiverse sequences that ideally fold to the same structure that improve the binding affinity.
 - Since ADFLIP cannot update the structure only when the model is incorrect in its sequence prediction, can it improve the binding affinity.
 - The benchmarks here also don't make sense, as why use TM-score rather than comparing the raw structure RMSD with Chai-1 for example, as done in prior evaluations.
 - Furthermore, the affinity gain, which is defined as whether the generated sequence’s predicted binding affinity exceeds that of the wild-type sequence, shows that in 42% of the time the generated sequences get worse than the reference's.

Overall, the metrics are quite soft, and while an interesting approach, leaves more questions. Also a general a 10% improvement over the wild-type binding affinities may also be more difficult and not physically useful due to the exponetial relationship between concentration and affinity. Unclear how this 10% factors into the problem as currently written though.

**Other Comments Or Suggestions:**

nit: FLow in line 24

**Other Strengths And Weaknesses:**

The paper is missing key numerical details with regards to how the model was training, specific hyperparameters for training and generation. Key ablations like why use purity sampling and what happens when using standard DFM sampling are needed to bostler the need for the technical novelty.

**Questions For Authors:**

1. What is more impactful, the architecture introduced or the discrete flow matching? A lot of attention to the architecture is provided and it clearly is beneficial but deeper ablations as to what degree does the underlying generative framework and architecture play a role would strengthen the contribution.
2. What are the mean/std or distributions of the sequence recoveries?
3. What happens when you do not use purity sampling wrt the benchmarks?
4. How is classifier guidance implemented for the discrete flow models? A algorithm would be useful here.

**Relation To Broader Scientific Literature:**

ADFLIP enables all-atom inverse folding with SOTA sequence recovery rates for protein complexes with ligands, nucleotides or metal ions. This is an important step for improving AI assisted protein design.

**Theoretical Claims:**

No theoretical claims.

The hyperparmaters used to sample as well as the foundation for guidance and specifics are not provided.

---

> ### Author Rebuttal · Authors · 2025-03-31
>
> **1.What is more impactful, the architecture introduced or the discrete flow matching? A lot of attention to the architecture is provided and it clearly is beneficial but deeper ablations as to what degree does the underlying generative framework and architecture play a role would strengthen the contribution.**
>
> Thank you for your comment. Both the multi-scale GNN architecture and the discrete flow matching framework play important and complementary roles in ADFLIP.
> The flow matching framework provides a flexible generative backbone that allows for integrating diverse sources of information—such as multiple structural states and external guidance signals. As shown in Table 4, this enables improved generation performance under dynamic structural contexts.
> On the other hand, the multi-scale architecture dynamically captures both atom-level and residue-level information, enabling the model to reason over partial side-chain context during sampling. To assess its effect, we conducted an ablation study:
> | **Model**              | **Ligand (%)**     | **Nucleotide (%)** | **Metal Ion (%)**  |
> |------------------------|--------------------|---------------------|---------------------|
> | ADFLIP w/ sidechain    | 62.19 ± 13.60       | 50.21 ± 13.52       | 75.79 ± 18.18       |
> | ADFLIP w/o sidechain   | 61.43 ± 16.20       | 49.74 ± 13.32       | 75.92 ± 16.52       |
>
> While the numerical gains in recovery rate are moderate, we believe the use of side-chain information has significant implications for downstream applications of inverse folding, such as enzyme design or protein–ligand interaction modeling—because the chemical interactions with small-molecule ligands or substrates are typically through the protein side chains, and where even minor torsional differences in side chains can result in substantial functional changes. We will point out this advantage of our all-atom approach more explicitly in the revised version.
>
> **What are the mean/std or distributions of the sequence recoveries?**
> We have computed the mean and standard deviation of the sequence recovery rates for LigandMPNN and ADFLIP as below. We will also include the distribution in the revised manuscript.
> | Method     | Ligand (%)        | Nucleotide (%)    | Metal Ion (%)     |
> |------------|-------------------|-------------------|-------------------|
> | LigandMPNN | 57.96 ± 11.77     | 46.14 ± 12.13     | 69.31 ± 17.46     |
> | ADFLIP     | 62.19 ± 13.60     | 50.21 ± 13.52     | 75.79 ± 18.18     |
>
>
>
> **What happens when you do not use purity sampling wrt the benchmarks?**
> We performed an ablation study to evaluate the impact of purity sampling on performance across benchmarks. Specifically, we varied the purity threshold and compared it to a fixed-step denoising strategy.
> Purity Sampling (variable threshold):
> | Threshold | Average Non-Protein RR | Std Dev |
> |-----------|------------------------|---------|
> | 0.3       | 0.602                 | 0.1399  |
> | 0.5       | 0.603                 | 0.1411  |
> | 0.7       | 0.588                 | 0.1457  |
> | 0.9       | 0.568                 | 0.1459  |
>
> Fixed-Step Sampling:
> | Denoise Steps | Average Non-Protein RR | Std Dev |
> |---------------|------------------------|---------|
> | 2             | 0.565                 | 0.1313  |
> | 5             | 0.592                 | 0.1350  |
> | 10            | 0.593                 | 0.1361  |
>
> We will add these new results as supplementary data to the revised manuscript.
> [Will we add these to the revised manuscript?] Kai: Yes
>
>
>
> **How is classifier guidance implemented for the discrete flow models? A algorithm would be useful here.**
>
> Thank you for this suggestion. We have included the training-free classifier guidance sampling algorithm below, which we will incorporate into the revised manuscript for clarity.
> ### Algorithm: Training-Free Classifier Guidance Sampling
>
> **Input:**
> - $N$ protein and non-protein structures $\{x_1, ..., x_N\}$
> - Initial sequence $s_0 = (\texttt{[MASK]}, ..., \texttt{[MASK]})$
> - Initial sidechains $\chi_0 = \emptyset$
> - Time step $\Delta t$
> - Denoising network $f_\theta$, sidechain packing network $g_\eta$, regressor network $h_\phi$
> - Target property value $y$
>
> **Procedure:**
> 1. Initialize $t = 0$
> 2. **While** $t < 1$:
>    - **For** $n = 1$ to $N$:
>      - Compute $p^n(\hat{s}_1) = f(s_t, x_n, \chi_t)$
>    - Average over structures:
>      $p(\hat{s}_1) = \frac{1}{N} \sum_n p^n(\hat{s}_1)$
>    - Compute predicted property:
>      $\hat{y} = h_\phi(p(\hat{s}_1))$
>    - Compute guidance respect to $s_1$:
>        $$p(y \mid \hat{s}_1) \approx -\nabla||y - \hat{y} ||^2$$
>    - Sample $s_1 \sim p(\hat{s}_1) \cdot p(y | \hat{s}_1)$
>    - **For** $n = 1$ to $N$:
>      - Compute sidechains:
>        $\chi^n_1 = g_\eta(s_1, x_n)$
>    - Compute reward:
>      $R_t(s_t, j) = E_{p_{1|t}(s_1|s_t)}[R_t(s_t, j|s_1)]$
>    - Sample $s_{t+\Delta t}$ using Eq.1
>    - Update time: $t \leftarrow t + \Delta t$
> 3. **Return**: Final sequence $s_1$

---

> > ### Comment · Reviewer_dLqF · 2025-04-03
> >
> > Thank you for answering my questions. I maintain my score.

---

### Official Review · Reviewer_BnUA · 2025-03-14

**Overall Recommendation:** 3

**Summary:**

A method named ADFLIP is proposed for inverse folding in all-atom structural contexts, e.g., containing ligand, nucleotide, and metal ions. The method is based on conditional discrete flow matching and a hierarchical GNN architecture. Additionally, it incorporates amino acid sidechains predicted by an external model as context and is able to perform training-free classifier guidance. Experimental results show improved performance over a re-trained version of LigandMPNN for diverse recovery metrics. The authors also show increased affinity gain measured in silico when an external model, DSMBind, is used for guidance.

## update after rebuttal

After the rebuttal, the authors addressed some of my concerns and I have raised my score.

**Claims And Evidence:**

The claims made in the manuscript are supported by clear evidence.

**Essential References Not Discussed:**

References for Inverse Folding such as [REF1] using discrete diffusion are missing.
[REF1] Yi, Kai, et al. "Graph denoising diffusion for inverse protein folding." Advances in Neural Information Processing Systems 36 (2023): 10238-10257.

**Experimental Designs Or Analyses:**

The experimental designs and analyses seem valid. Re-training LigandMPNN for the proposed cluster might affect the fairness of evaluation depending on the rigorousness of the checkpoint chosen for evaluation.

**Methods And Evaluation Criteria:**

The evaluation criteria make sense for the problem investigated.

**Other Comments Or Suggestions:**

1. (Line 97) Typo: “Saport”
2. (Line 77-78) The word “here” appears twice.
3. The equation numbering appears to be wrong throughout the manuscript.
4. Algorithm 1 appears before Fig. 1 in the text. Might need to re-arrange the order.
5. (Line 413) Typo: “DSMBIND”

**Other Strengths And Weaknesses:**

Strengths:
2. A discrete flow matching framework is proposed for all-atom protein sequence design. As the denoiser network, the authors combine residue-wise and atom-wise features obtained by a GNN, and a transformer-based architecture is used for decoding.
3. The authors propose the use of an external network to sample sidechains during the sequence decoding process to provide additional information in all-atom contexts when decoding.
1. The proposed method is able to handle multiple conformations, and the adaptive sampling proposed by the authors might be useful for other inverse folding algorithms.

Weaknesses:
1. The manuscript would improve with additional information and clarification about the methodology.
2. Additional ablation studies regarding sequences generated by the model and how different components affect the overall performance seem needed.
3. The methodology for the guidance by binding affinity using the fixed input structure and DSMBind is arguable, even though it shows the ability of the proposed method for guided generation.

**Questions For Authors:**

1. (Related Work) The reviewer suggests re-writing the Related Work section. Additionally, references for Inverse Folding like [REF1] using discrete diffusion are missing.
[REF1] Yi, Kai, et al. "Graph denoising diffusion for inverse protein folding." Advances in Neural Information Processing Systems 36 (2023): 10238-10257.
2. (Methodology Explanation) Figures 1/2 and the methodology section writing do not give enough information for the reader to understand the steps of the conditional flow matching methodology. Specifically, from my understanding, the flow matching is generating the distribution s_t at each timestep, more information or illustrations regarding this process seems needed.
3. (Classifier Guidance) Does the training-free classifier guidance methodology work in a similar fashion to potential in other generative protein models like Chroma and RFDiffusion? What are the differences between these approaches and your flow matching-based approach?
4. (Methodology Clarification) Removing the side chains for masked positions might still influence the decoding toward sequences that were sampled at the beginning of the denoising process. Do the authors discuss or have an experiment to test this factor or how it influences the generation? An ablation study would help the effect of using the sidechain context.
5. (Need for Ablation Studies) What is the influence on the results of adding the sidechains? What is the influence on the results of using only the GNN architecture with a sequence decoder, in this case, is your architecture similar to the current LigandMPNN formulation?
6. (Conditional Flow Matching Formulation) Additional information about the conditional and the formulation from Campbell et al would be helpful for readers.

**Relation To Broader Scientific Literature:**

The key contributions of the manuscript are related to:

1. Proposing a conditional flow matching framework for all-atom protein sequence design.
2. Incorporate partial side-chains using a pre-trained model during the sequence reconstruction process.

**Theoretical Claims:**

The theoretical claims seem correct. Additional information about the conditional discrete flow matching formulation in (Campbell et al, 2024) would help to verify the claims in the manuscript.

---

> ### Author Rebuttal · Authors · 2025-03-31
>
> **1.Related Work**
>
> Thank you for pointing this out. We mentioned the paper by Yi et al. [1] in the introduction, but agree it should also be included in the related work. We will rewrite the related work section and discuss the discrete diffusion method for inverse protein folding from Yi et al and also another diffusion model [2][3]. This will help clarify how our work connects to previous studies.
>
> [1]Yi, Kai, et al. "Graph denoising diffusion for inverse protein folding." Advances in Neural Information Processing Systems 36 (2023): 10238-10257.
>
> [2]Wang, Xinyou, et al. "Diffusion language models are versatile protein learners." arXiv preprint arXiv:2402.18567 (2024).
>
> [3]Wang, Xinyou, et al. "Dplm-2: A multimodal diffusion protein language model." arXiv preprint arXiv:2410.13782 (2024).
>
>
> **2.Methodology Explanation for Flow Matching**
>
> Thank you for this suggestion. It is correct that, in our framework, discrete flow matching generates the distribution $p(s_t​)$ at each timestep. However, unlike continuous flow models, where the denoiser predicts noise, the denoiser of discrete flow matching directly predicts data (e.g., sequence tokens).
> In our implementation, a trained denoiser first estimates $p(s_1)$, and then $p(s_t)$ is computed by taking an expectation over this estimate, as shown in Equation (line 137) and Algorithm 1.
>
> In the revised manuscript, we will add clearer descriptions, expanded mathematical formulations, and improved illustrations to explain the conditional discrete flow matching process more intuitively.
>
> **3.Classifier Guidance**
>
> Our classifier-guidance approach shares a similar objective with Chroma and RFdiffusion in that we aim to condition generation on an external property $y$, estimated via $p(y∣s_t)$. However, there is a key difference in how this is achieved.
> Chroma and RFdiffusion use continuous diffusion models and typically require training a separate classifier or regressor (y = f(s_t)) directly on noisy intermediate states, $s_t$. In contrast, our method uses a training-free guidance strategy: we leverage a pre-trained classifier or regressor that operates on clean data $s_1​$, such as AlphaFold.
> We estimate $p(y∣s_1)$ from this clean output and then derive $p(y∣s_t)$ by taking the expectation over s1∼p(s_1∣s_t), as described in Equation (line 307). This approach avoids the need to retrain an external model and enables seamless integration of existing structure-based predictors.
>
> **4.Methodology Clarification: Does removing sidechains at masked positions bias decoding toward early samples, and is there an ablation study on this effect?**
>
> Thank you for this insightful comment. We conducted an ablation study on the parameter $\tau$ for purity sampling, which controls how much side-chain information is preserved for masked positions during the denoising process. A smaller $\tau$ value retains more side-chain atoms, providing richer structural context to the denoiser. Conversely, a higher $\tau$ results in fewer sampled residues and less side-chain information.
> | $\tau$ | Interaction Recovery Rate | Std. Dev. |
> |------------------------------------|-------------------------------|-----------|
> | 0.3                                | 0.602                         | 0.1399    |
> | 0.5                                | 0.603                         | 0.1411    |
> | 0.7                                | 0.588                         | 0.1457    |
> | 0.9                                | 0.568                         | 0.1459    |
>
> **5. Need for Ablation Studies for sidechain**
>
> We conducted an ablation study to assess the impact of incorporating predicted side-chain atoms into the denoiser's input. The results are summarized below:
> | Model               | Ligand (%)        | Nucleotide (%)    | Metal Ion (%)     |
> |---------------------|-------------------|-------------------|-------------------|
> | ADFLIP w/ sidechain | 62.19 ± 13.60     | 50.21 ± 13.52     | 75.79 ± 18.18     |
> | ADFLIP w/o sidechain| 61.43 ± 16.20     | 49.74 ± 13.32     | 75.92 ± 16.52     |
>
> While the inclusion of side-chain information leads to moderate improvements in sequence recovery, we believe its primary value lies in enhancing biological fidelity rather than optimizing this metric alone. In many downstream tasks—such as enzyme design or protein–ligand interaction modeling—minor torsional differences in side chains can lead to substantial functional changes (e.g., in binding affinity or catalytic activity). Therefore, even small gains in sequence recovery reflect a more meaningful structural signal that allows the model to better capture fine-grained biophysical nuances. We will discuss this in more detail in the revised manuscript.
>
> **6.(Conditional Flow Matching Formulation) Additional information about the conditional and the formulation from Campbell et al would be helpful for readers.**
>
> We agree, and will add more detail about Conditional Flow Matching in the revised manuscript.

---

> > ### Comment · Reviewer_BnUA · 2025-04-03
> >
> > Thank you for answering my questions.
> >
> > I have raised my score.
> >
> > I have a couple additional comments/questions:
> >
> > 1. I think these ablation studies for sidechains and the effect of removing sidechains should be incorporated as part of the manuscript and discussed further.
> >
> > 2. The quality of the denoiser directly affects the training-free classifier guidance. Additionally, the classifier might have a different objective (improve functionality) compared to inverse folding. In addition to the foldability score, it would be important to check the RMSD between for the predictions for these cases as the classifier might be also leading to mutations that change the structure.

---

### Official Review · Reviewer_1WvW · 2025-03-14

**Overall Recommendation:** 3

**Summary:**

This paper proposed a new method, namely ADFLIP, a generative model for inverse protein folding that designs sequences based on all-atom structural contexts. It is designed to handle complexes with non-protein components and dynamic structures using ensemble sampling. ADFLIP progressively incorporates side-chain context and leverages classifier guidance sampling for sequence optimization. The authors showed experimental results on a dataset curated from PDB.

**Claims And Evidence:**

- The main claim of this paper is supported by evidence of experimental results.

- The results are shown for only one dataset of inverse folding. I understand this dataset used here is more aligned with the claim, however, the question naturally comes to mind how this method would perform compared to other method in the more widely used in inverse folding datasets such as CATH 4.2 and CATH 4.3.

- The comparison is shown against PiFold, ProteinMPNN, and LigandFold. However, there are other more recent methods that significantly outperform PiFold and ProteinMPNN on other dataset such as CATH 4.2, CATH 4.3, TS50, TS500, etc. Some examples include LM-Design (Zheng et al. 2023), DPLM (Wang et al. 2024), AIDO.Protein (Ning et al. 2024), and so on. Although none of these methods use use all-atom structural context or the ligands, I would how ADFLIP would perform against those methods in their evaluation dataset as well as the most traditional ones.

**Essential References Not Discussed:**

N/A

**Experimental Designs Or Analyses:**

- The experimental design and analyses are valid.

- I appreciate how the authors not only showed the recovery rate and perplexity, but also foldabity as well as related metrics such as TM-score, pLDDT, and RMSD.

**Methods And Evaluation Criteria:**

- The method is sound and properly described.

- The evaluation criteria is also sound. However, there is some gap in the evaluation on more widely used dataset and against more recent methods.

**Other Comments Or Suggestions:**

N/A

**Other Strengths And Weaknesses:**

N/A

**Questions For Authors:**

N/A

**Relation To Broader Scientific Literature:**

The key contribution of this paper is related to proteomics research as well as machine learning research with such biological data. Their provided method has good use-case in applications such as drug design and discovering therapeutics.

**Theoretical Claims:**

- The authors provided clear mathematical derivation of their proposed approach

- The demonstration algorithms are also sound and properly explained.

---

> ### Author Rebuttal · Authors · 2025-04-01
>
> **How does ADFlip perform on protein only dataset such as CATH**
>
> Thank you for the suggestion. We retrained ADFlip on the CATH 4.2 dataset and evaluated its performance based on sequence recovery rate. The results are summarized below:
>
> | Method        | Sequence RR (%) |
> |---------------|-----------------------------|
> | ProteinMPNN   | 45.96                       |
> | PiFold        | 51.66                       |
> | **ADFlip**    | **52.13**                   |
>
> **How does other inverse folding model perform on all-atom dataset**
>
> Thank you for the suggestion. Due to time constraints during the first review period, we are still working on additional evaluations. We aim to include results for other inverse folding models on the all-atom dataset in the second-round discussion and the revised manuscript.

---

### Official Review · Reviewer_kagU · 2025-03-20

**Overall Recommendation:** 1

**Summary:**

This paper introduces ADFLIP, a model for inverse protein folding designed for complex biomolecular systems. By by incorporating all-atom structural context (including protein backbone, non-protein components like ligands and metal ions, and progressively predicted side chains) and handling dynamic protein complexes with multiple structural states.

**Claims And Evidence:**

The paper heavily emphasizes "all-atom" in the title and abstract, implying it's a novel and crucial advantage. However, the meaning of "all-atom" in the context of ADFLIP and its distinctiveness from other methods are not clearly and convincingly established. PiFold and other GNN-based methods do use geometric features constructed from all atoms (backbone atoms) to represent protein structure. The geometric features (distances, angles, etc.) inherently rely on the coordinates of all atoms. Therefore, simply stating "all-atom" as an advantage is misleading because it's not a feature unique to ADFLIP compared to modern GNN-based methods. The input is more expressive, it should be distinct from inverse folding problem based on backbone atoms.

**Essential References Not Discussed:**

The problem setting can be solved by the method proposed by UniIF [1] published in NeurIPS 2024.

[1] Gao, Zhangyang, et al. "Uniif: Unified molecule inverse folding." Advances in Neural Information Processing Systems 37 (2024): 135843-135860.

**Experimental Designs Or Analyses:**

The central claim regarding "all-atom" as a unique and clearly defined advantage is weakly supported and potentially misleading due to the lack of clear definition, lack of isolation of the "all-atom" effect, and potentially inaccurate characterization of baselines.

**Methods And Evaluation Criteria:**

The proposed methods in ADFLIP, particularly discrete flow matching, all-atom context awareness, GNN architecture, and ensemble handling, are make sense for the problem of inverse protein folding in complex biomolecular systems. However, I fail to recognize the key technical innovations.

**Other Comments Or Suggestions:**

N/A

**Other Strengths And Weaknesses:**

This paper is well-organized and clearly-written. But the problem setting is quite simple. It seems to extend the traditional inverse folding into protein complex scenarios.

**Questions For Authors:**

Could you please clearly define what "all-atom" specifically means in the context of ADFLIP and how your approach to "all-atom" context is fundamentally different and uniquely advantageous compared to how baseline methods utilize all-atom geometric information? Simply considering non-protein components and side-chain prediction doesn't seem to be a fundamentally different "all-atom" concept.

**Relation To Broader Scientific Literature:**

It is related to inverse folding problem, which is an important area in molecular biology.

**Theoretical Claims:**

N/A

---

> ### Author Rebuttal · Authors · 2025-03-31
>
> ## Clarification on "All-Atom" Structure in ADFLIP
> Perhaps we had not explained clearly enough what we mean by ‘all-atom’.   This interpretation of “all-atom” was introduced previously with RoseTTAFold-all atom[1].
> In this context, all-atom refers not only to including the full set of protein atoms (backbone and side chains) but also to incorporating non-protein biological components such as ligands, ions, and nucleotides. Most inverse folding methods—including PiFold, ProteinMPNN, and others—limit their input representation to backbone atoms (typically N, C, O).
>
> By contrast, our method, ADFLIP, is designed to operate on full biological assemblies that may include a broader set of atoms from diverse molecules beyond proteins. This enables modeling protein-ligand, protein-RNA, or protein-ion interactions in a generalizable manner. Therefore, our all-atom terminology emphasizes the inclusion of all relevant atomic context in biomolecular complexes—not just the geometric features derived from protein atoms—enabling applications in more complex and realistic biological environments.
> We will revise the introduction and related work to more clearly explain this distinction.
>
> ## Novelty and Technical Contributions of ADFLIP
> While previous works such as UniIF have explored inverse folding in all-atom settings, the objectives differ. UniIF is designed as a general framework for modeling all biological structures, including proteins, RNAs, and small molecules, rather than focusing specifically on protein design. In contrast, ADFlip is specific for protein design tasks, particularly for proteins that interact with ligands, nucleotides, and other biomolecules. Our work introduces several novel contributions:
> 1. Generative Modeling with Discrete Flow Matching:
> ADFLIP is, to our knowledge, the first inverse folding framework that applies *discrete flow matching* to protein sequence design, taking into account protein complex context. This is particularly advantageous when handling input structures with high uncertainty, which are common when flexible ligands or RNA components are present, where traditional autoregressive models struggle. Our experiments demonstrate that ADFLIP outperforms autoregressive baselines on both single-structure and multi-structure inputs.
> 2. Training-Free Classifier Guidance for Conditional Generation:
> We propose a novel, training-free classifier-guidance approach for conditional generation with flow matching. Existing guidance techniques often require retraining regressors or classifiers to accept noisy intermediate inputs $ s_t $. Instead, we use a pre-trained classifier/regressor  (e.g., AlphaFold) that operates on clean inputs $ s_1$, and approximate $ p(y | s_t)$ by taking the expectation over $ p(s_1 | s_t) $, as described in Eq. (line 308). This allows us to plug in any pretrained predictor *without* retraining, enabling flexible and efficient conditional design.
> 3. Support for Multiple Structural States:
> ADFLIP supports input ensembles with multiple structural conformations, capturing dynamic aspects of protein complexes. This is crucial for modeling biological systems where a single static structure may be insufficient.
> In summary, ADFLIP introduces a general-purpose, generative approach to inverse folding in full biological assemblies by (1) leveraging a broader all-atom context, (2) introducing discrete flow matching for improved sample quality and uncertainty modeling, and (3) enabling flexible, training-free guidance using pretrained models.
> We hope this response clarifies the distinctiveness of ADFLIP and addresses your concerns. We thank you for your valuable feedback, which will help in improving our manuscript.
>
> [1]Krishna, Rohith, et al. "Generalized biomolecular modeling and design with RoseTTAFold All-Atom." Science 384.6693 (2024): eadl2528.

---

### Decision · Program_Chairs · 2025-05-01

**Decision:**

Accept (poster)

**Comment:**

This paper presents a new generative model called ADFLIP for inverse protein folding based on all-atom structural contexts using discrete flow matching. ADFLIP progressively adds  predicted amino acid chains as structural context during sequence generation and allows pretrained models to guide the sequence generation process. Experimental results prove the effectiveness of the proposed approach over the current state-of-the-art approach LigandMPNN.

The proposed approach is novel and shows very competitive performance over LigandMPNN. The AC hence believes the proposed approach makes a good contribution for the protein design community and vote for acceptance. The authors should address some of the limitations in the revised version (e.g., evaluating the model on more than one data sets, comparison with more recent approaches).